# Brain-Directed Care: Why Neuroscience Principles Direct PICU Management beyond the ABCs

**DOI:** 10.3390/children9121938

**Published:** 2022-12-09

**Authors:** Debbie A. Long, Michaela Waak, Nicola N. Doherty, Belinda L. Dow

**Affiliations:** 1School of Nursing, Centre for Healthcare Transformation, Queensland University of Technology, Brisbane, QLD 4059, Australia; 2Paediatric Intensive Care Unit, Queensland Children’s Hospital, Brisbane, QLD 4101, Australia; 3Centre for Children’s Health Research, The University of Queensland, Brisbane, QLD 4101, Australia; 4Regional Trauma Network, SPPG, DOH, Belfast BT2 8BS, Northern Ireland, UK; 5School of Psychology, Faculty of Life and Health Sciences, Coleraine Campus, Ulster University, Coleraine BT52 1SA, Northern Ireland, UK

**Keywords:** critical care, pediatrics, child development, neuroscience, trauma informed care

## Abstract

Major advances in pediatric intensive care (PICU) have led to increased child survival. However, the long-term outcomes among these children following PICU discharge are a concern. Most children admitted to PICU are under five years of age, and the stressors of critical illness and necessary interventions can affect their ability to meet crucial developmental milestones. Understanding the neuroscience of brain development and vulnerability can inform PICU clinicians of new ways to enhance and support the care of these most vulnerable children and families. This review paper first explores the evidence-based neuroscience principles of brain development and vulnerability and the impact of illness and care on children’s brains and ultimately wellbeing. Implications for clinical practice and training are further discussed to help optimize brain health in children who are experiencing and surviving a critical illness or injury.

## 1. Introduction

Evidence of the short- and long-term effects of adversity on health and wellbeing in children is evolving, especially when adversity is persistent, cumulative, or occurs at critical junctures of neurodevelopment. Similarly, there is growing concern about the impact of hospitalization and critical illness and injury on child health and development. This potential threat to the present and future brain health of children, impacting their families and society, is motivating a shift in focus from the acute outcomes of the pediatric intensive care unit (PICU) to person-centered long-term outcomes. Our healthcare systems have evolved, with initial stages concentrating on acute and infectious diseases, hospitals and physicians, and single cause and effect relationships, as well as specific therapies. More recently, our emphasis has been on chronic illness and disability, expanded sub-specialization and technology, interdisciplinary person-centered care, and connections to the community. However, to provide a continuum of care that goes well beyond healthcare systems and includes lived experience and community partners, our systems and practices must implement population-based prevention and early intervention and prioritize optimal health status for everyone. To date, neuroscience principles and their implications for practice have not fully reached PICU research and practice. The purpose of this article is to highlight the significance of brain development and vulnerability for care pathways, with implications for PICU research, practice, and training.

## 2. Neuroscience: Building Health Brains

Current research is elucidating the processes associated with neurodevelopment and learning in children and into adulthood. Hypotheses have evolved from the concept of the brain’s growth following a fixed biological course, and its development being purely defined genetically, to understanding that most brain cells originate prior to birth and migrate into their pre-defined location, with most connections between cells occurring throughout infancy and childhood, with some ongoing synaptogenesis occurring in young adulthood [1,2]. Recently, neuronal networks and “plasticity” have been proposed with evidence that early positive but also sensory experiences in infancy—vision, hearing, smell, touch, and taste—have an impact on how brains develop. A significant proportion of brain growth occurs antenatally, and by the age of two, a child’s brain has reached around 80% of its adult size. Our neurons are interconnected to create additional billions of unique brain circuits. Every time we have a novel experience, our brains use a new neural route [1]. Learning is the process by which new experiences lead to sustained new behaviors. If the experience is repeated or the stimulus is especially powerful, more nerve impulses are delivered via the new channel, with pruning occurring if new connections are not re-enforced. This improves learning and explains why repetition helps in acquiring new knowledge, by facilitating circuits within a network [3]. Hardwiring is the term used to describe this process, and by the age of three, 90% of the connections will be made, demonstrating the persistent effects of early experiences.

### 2.1. Brains Are Built, Not Born

Neural networks (our brain architecture) are built by way of a continuous process that starts before birth and lasts till maturity. The strength or fragility of these networks during crucial timepoints of neurodevelopment determine the stability and quality of the development of all subsequent abilities and behaviors. As the brain develops in a hierarchical “bottom-up” fashion, skills build upon themselves [4]. Over time, simpler circuits and networks are built upon to create increasingly complex circuits and skills. As the brain matures, its propensity for plasticity declines and its circuits become more stable, making later changes much more difficult. Circuits that are not being used are continually pruned, consistent with the concept of developmental windows or critical periods [5,6]. Learning, behavior, health, and wellbeing are all closely related throughout the course of life. Physical, cognitive, emotional, and social abilities are intrinsically linked to one another; positive or negative developments in one domain will have an impact on all others [7].

### 2.2. Experiences Shape Our Brains

There is increasing evidence that early experiences and interaction are key to developing strong brain architecture [4,8]. Environmental factors enhance how functional pathways are utilized and impact on structural brain development [9]. These phases of development are known as sensitive periods because certain experiences can either increase or decrease brain connections [4,10]. Our perceptual, cognitive, and emotional capacities are all established on the foundation of our early experiences. Furthermore, if the presence or absence of an experience leads in permanent change, this time period may be referred to as ‘critical’ [11]. 

Early life stressors influence the development of neurobiological systems responsible for health and wellbeing across the lifespan [12]. Stress may be especially detrimental during early rapid brain development, skewing ongoing formation of brain pathways and structure towards vulnerability. Studies on adverse childhood experiences (ACEs) have informed how toxic stress can impact child development and health and wellbeing into adulthood, including chronic physical, mental and social health problems [13,14,15,16]. Despite being less researched, exposure to counter-ACEs—positive childhood experiences—seems to protect against ill health and to foster improved health and wellbeing throughout adulthood [17]. Additionally, counter-ACEs are linked to improved executive function, a greater internal locus of control, and stronger attachments to one’s family as adults, which protects against depression, stress, and sleep problems [18,19,20,21,22,23]. However, research using brain imaging to examine the impact of early stress on the brain has mostly targeted older children and adults [24,25]. Additionally, there are gaps in our knowledge of how less severe sources of personal, parental, or familial stress affect early development, as early life stress has frequently concentrated on more extreme sources of adversity [26]. Animal studies have demonstrated that mild prenatal stress (e.g., noise pollution) can cause emotional and brain structural modifications, delays in feeding, distractibility and motor delays in offspring [27]. Large international studies have also demonstrated that milder transitory stressors (light and noise pollution) impair reading comprehension and recognition, information recall and conceptual recall memory in primary school children [28,29,30,31]. 

New research elucidating how stress can cause brain dysfunction through hormonal and immune dysregulation and neuro-inflammation might offer not only pathophysiological insights but also potential treatment targets [32]. Although inflammation can be brought on by other experiences, such as seeing traumatic events or going through stressful situations, it is also an essential component of an immune system’s response to invaders and threats. This response aids our bodies in responding to and surviving those experiences [33]. However, prolonged periods of stress can lead to harmful hormonal and inflammatory dysregulation (See Figure 1). This state of imbalance and constant activation during critical periods of development can cause organs to adapt in ways that affect all ages, increasing their chance of developing conditions including diabetes, asthma, depression, and dementia as well as cardiovascular disease and other chronic diseases [33].

### 2.3. The Importance of Human Relationships

Building healthy brain architecture through nurturing and responsive connections lays a strong foundation for learning, behavior, and health [34,35,36,37]. When relationships are interrupted in some way, levels of stress hormones increase, and dysregulation of the immune system can contribute to brain inflammation, affecting the structure of the brain and preventing the development of healthy neural pathways. The circuitry of the developing brain is shaped by the interaction of genes and experience. Young children frequently offer invitations to interact with adults who either respond to their needs or not. The wiring of the brain is fundamentally based on this “serve and return” mechanism, especially in the early years [38]. Young children instinctively try to engage others in conversation by vocalizing, making facial expressions, and using gestures. Unfortunately, the child’s learning process is interrupted and may have detrimental effects on subsequent development, if adults are unable or impeded from responding in a complementary manner [39]. 

### 2.4. Resilience Is Fundamental for Fostering and Advancing Neurodevelopment

Resilience is the ability to flexibly adapt to changing conditions and to recalibrate in the context of challenge or adverse experiences [40]. Resilience develops over time as gene expression and epigenetics are shaped by experiences, especially during sensitive stages of development. The individual traits that allow adaptive or maladaptive behaviors depend on the individual’s neurological capacity [41,42,43]. Executive function, also referred to as the brain’s air traffic control system, is strongly influenced by key interactions at an early age [44,45]. Children exposed to early life stress therefore could be at increased risk, as recent research indicates that executive function abilities, rather than intellect, predict resilient school and peer functioning [46,47]. 

## 3. Impact of Hospitalization and Critical Illness on Child Health

It is now understood that hospitalization in early childhood, chronic illness or a life-threatening illness or injury can be a negative experience that challenges crucial brain development and strains the family system, placing a child at risk of adverse and lasting effects on their brain development and ultimately on health and well-being [48,49]. The negative experience can include not just the illness or the injury, but also aspects of the hospitalization and treatments provided. These aspects can be experienced in different ways, but impact on all areas of neurodevelopment [50,51,52]. Using neuroscience principles on how to support healthy brain development, we can also apply these strategies for protecting brains—particularly those that are still developing. 

### 3.1. Hospitalization in Childhood 

Improved understanding of the psychological and neurological impacts of pediatric hospitalizations requires a developmental perspective. Hospitalization can result in a disruption in familial and social relationships and routines, which can be experienced as a stressful, traumatic event in a person’s life [52]. This process is especially intensified in early childhood, given developing brains are more vulnerable and have had less opportunity to develop protective resilience and coping mechanisms. In addition to separation from carers, the use of invasive procedures, pain, impacts of medications and sensory overstimulation amplify each other’s negative effects [53,54,55,56,57]. In short, hospitalization can act as a developmental and attachment interrupter.

Early studies have suggested that Children under the age of four are more likely than older children to experience both short- and long-term unfavorable effects from hospitalization [58,59,60,61]. Through more therapeutic hospital experiences, neurodevelopmental resilience factors appear to allow children from age 6 years onwards to develop coping strategies [62], thereby protecting their brains against stresses related to physical illness and its treatment [60]. Hospitalization in the first 48 months of life has been linked to an increased likelihood of psychiatric, internalizing, and externalizing disorders, and developmental vulnerability at primary school age [63]. Furthermore, multiple hospitalizations during early childhood suggest a dose response relationship to these adverse outcomes.

### 3.2. Chronic Illness

Chronic illnesses are physical health conditions that are protracted, complex to treat, and often associated with impairment or disability. When experienced in early childhood, chronic illnesses have the potential to profoundly influence a child’s developmental trajectory over and above the pure impact from the specific disorder [64,65]. According to research studying the link between academic achievements and child health, children with chronic illnesses have worse school outcomes than their healthy counterparts. Although the direct impact of chronic illness on neurodevelopment is harder to elucidate, there is evidence that factors including rising absenteeism and disengagement from school significantly contribute [66]. A small number of studies in younger children with special health care needs have found that they are also at risk of worse cognitive and psychosocial outcomes compared to healthy peers, further suggesting that illness in early childhood may influence the development of skills that are crucial for academic success [67]. Another study found that chronic illness in young children was a risk factor for reduced school readiness across multiple domains, regardless of the number or type of conditions [68]. Of note, this cohort included, but was not limited to, common childhood illness such as chronic otitis media and asthma, in addition to more complex chronic illnesses such as epilepsy and musculoskeletal conditions. It is hypothesized that the cumulative stress of repeated exposure to procedures and attendance in health care settings impacts brain development and decreases resilience, and a child’s negative developmental trajectory is escalated by increasing separation of their developmental trajectile from their peers.

### 3.3. Critical Illness

The pediatric intensive care unit (PICU) literature reflects a shift of focus from short term outcomes to what we now know as ‘Post-Intensive Care Syndrome in pediatrics’ (PICSp), which encompasses the four domains of cognitive, physical, socioemotional and communication impairments [69]. Whilst this evidence is still evolving, the current summary of work suggests an incidence of up to approximately 30% in PICU survivors [54,70,71,72,73,74,75,76,77,78,79,80]. Retrospective studies exploring the impact of a PICU admission show a significantly higher incidence of mental health diagnoses and psychotropic medication use in survivors [81], particularly in those with respiratory illness requiring invasive mechanical ventilation. Invasive mechanical ventilation is one of the life-saving therapies utilized in PICU, however their necessity and application are inexplicitly linked with the child’s underlying health, severity of illness, and treatments received. This has been confirmed in a systematic review assessing burden of mental illness in children admitted to PICU [73]. A population-based study of over 5000 children admitted to PICU prior to their 5th birthday also found that 14% of survivors did not meet the national minimum standard for Grade 3 educational achievement. PICU survivors performed significantly lower than matched controls across each domain of the national assessment, with socioeconomic status emerging as a strong non-disease related predictor of academic outcomes. This further raises the importance of additional support for children from socially disadvantaged families [82].

A prospective study in the neonatal intensive care unit (NICU) setting showed the impact of cumulative stressors during a NICU admission on connectivity on functional magnetic resonance imaging and motor function in clinical testing despite adjustment for disease severity. Whilst this is yet to be repeated in the PICU setting, it supports the hypothesis of adverse impacts of ICU related stressors on the developing brain over and above the effect of the underlying illness [83]. This has been underpinned with hypothesis generating literature linking stress during vulnerable periods of brain development to negative neurodevelopmental outcomes [84]. Importantly, strategies to mitigate post-PICU discharge morbidity should begin within the walls of PICU and continue after discharge.

## 4. Approaches to Reducing the Burden of Adverse Childhood Hospitalization and Illness Experiences

### 4.1. Trauma-Informed Care

Trauma-informed care (TIC) for has been adopted across a variety of service systems, including schools and health care [13]. Trauma, in this context, refers to a psychological or emotional experience rather than physical injury. Both children and their parents may experience heightened traumatic stress within and after discharge from the PICU, with up to 25% going on to develop PTSD (Post Traumatic Stress Disorder) [85]. Traumatic events may be the direct cause of PICU admission, such as an accidental injury, they might be more distant in a patient’s history, or as is now increasingly understood, the PICU admission itself may be traumatizing for children and families. 

Trauma-informed healthcare is an approach to caregiving based on the understanding of the link between trauma exposure and poor health outcomes and applies this knowledge about trauma to practice and policy to prevent re-traumatizing the patient through their healthcare experience and reduce negative sequelae [86]. A healthcare system that offers pediatric TIC (a) acknowledges the potentially traumatic nature of medical care, and (b) takes this understanding into account in organizational culture, policies, and procedures, as well as in every interaction between children, families and the healthcare team [86]. Incorporating an understanding of the patient’s behavior, considering current or previous trauma exposure, and aiming to provide supportive care that promotes the child and family’s feelings of safety and security are key principles [87,88]. Considering the parent–child dyad of pediatric acute care, parental inclusion can enhance TIC [89,90].

Every healthcare visit provides an opportunity to promote the benefits of family resilience and relational health [91]. Early relational health is the formation of strong foundational relationships in the first three years of life, which are essential for a child’s successful physical, emotional, and moral development [92]. In a broader sense, relational health is relevant to all age groups, is dyadic, and encompasses both the child’s and caregiver’s capacity to establish a secure, supportive connection that enables both to thrive. These connections foster resilience and serve as a stress buffer for children, making them an important first line of defense against stress-related problems [93]. Effective implementation of trauma-informed care will require changes at the individual level and in organizational policies and procedures [88,94]. It will also likely require a commitment from organizational leadership to educate all staff who engage with children in specific new skills and understandings (e.g., recognizing the psychological impact of medical events and treatment on children, providing effective support for children during challenging treatment experiences, and helping parents support children throughout their hospital stay) [94,95]. Even brief training has been shown to increase clinicians’ knowledge and confidence in implementing trauma-informed practices in their daily interactions with children [96]. Tools, including brief, focused, online training resources, are available to help interdisciplinary teams learn and implement the specific skills required for trauma-informed pediatric healthcare [97,98,99]. Research demonstrating the link between trauma and reduced child–parent attachment provides a strong rationale for including ACE measures into comprehensive screening and intervention with families [100,101]. Given these vulnerable families may also be disadvantaged, routine screening and support for social determinants of health in families of critically ill children should also be considered [102]. Demers et al. provide a comprehensive discussion on the principles and evidence for TIC in PICU [103]. 

### 4.2. PICU Liberation

Evidence-based bundles such as the ‘A–F’ bundle or PICU Liberation incorporate concepts of neuroscience and neurorehabilitation into daily patient cares to optimize early recovery. Each bundle element (***A***ssess, prevent, and manage pain, ***B***oth spontaneous awakening and breathing trials, ***C***hoice of analgesia and sedation, ***D***elirium: assess, prevent and manage, ***E***arly mobility and exercise, and ***F***amily engagement and empowerment) addresses aspects of care that can prevent PICU-related complications and achieve brain protection by means of reduction in duration of invasive procedures required as well as interventions offered to allow neurodevelopmentally appropriate cares to occur [104]. Some bundle elements are informed by neurodevelopmental care principles, which aim to: (1) reduce pain and stress, (2) encourage and support parents in their primary caregiver role, (3) create an environment that facilitates healing, (4) safeguards daily routines, and (5) supports family-centered care [105]. These practice changes are often nurse-led and require an interdisciplinary team allowing additional interventions tailored to the child’s and family’s needs. Play has been shown to have a high therapeutic value for sick children while they are in the hospital, helping to promote their physical and mental well-being as well as their recovery [106]. Play has been widely used to reduced pain and anxiety related to invasive procedures and chemotherapy, and aids in exploring concerns relating to the child’s hospital experiences and reduces the severity of the unpleasant emotions that follow a hospital admission [107]. Play can be incorporated into cognitive stimulation for delirium prevention, early mobilization and family engagement and encourages strong relationships with caregivers and clinicians, with serve and return interactions and the development of coping strategies.

Adult liberation data support improved short- and long-term outcomes if A–F bundles are implemented, demonstrating a strong dose response between elements implemented and outcome [108,109]. Similar rigorous data in pediatrics are still developing [110,111], although it has been hypothesized that impacts could be larger given potential exponential impacts on not only the developing brain but also the family unit. In early work which included 600 children, Lin et al. [112] demonstrated that every 10% increase in day specific PICU Liberation compliance correlated with a 1.4% reduction in mortality. Other smaller studies have also demonstrated reductions in the prevalence of delirium, length of mechanical ventilation, and PICU and hospital length of stay, and improvements in functional outcomes [113,114].

### 4.3. Parental Education and Psychosocial Support

Having a child admitted to PICU is associated with increased parental stress due to alterations in the parental role, procedures and their child’s appearance [115], as well as anxiety [116], depression and post-traumatic stress [117]. A child’s physical, social, emotional, and cognitive development is optimized when parenting is supportive and sensitive to their individual needs [118]. Unfortunately, a life-threatening illness and admission to the PICU can cause parental distress, which can have detrimental short- and long-term impacts on the parent and has been linked to children’s longer-term psychological issues [117,119]. Chronic stress could lead to impatient or strict parenting and impact on a parent’s ability to respond in constructive ways to their child’s ever-changing needs and functional limitations [120]. Optimal parenting skills during this critical period of development and vulnerability are vital for protecting and building healthy brains [121,122,123,124]; however, increased parental stress can limit their capacity to care and advocate successfully for their child. These stressors can increase during the transition from hospital to home and are frequently complicated by the child’s functioning, and the parent’s resilience, support, and knowledge [125,126,127,128]. Lack of support and continuity of care after PICU discharge can also intensify unmet needs, poor child outcomes, and parent wellbeing [129,130].

Approaches to parental education and support are emerging. Conceptually, they may follow a stepped-care approach from light-touch universal strategies to buffer distress and promote the child and family’s resilience, to more intense targeted prevention or intervention programs specifically aimed at children and families identified as high risk (via screening, identification of pre-existing vulnerabilities or identification of other known risk factors such as low socioeconomic status). Improving parental self-efficacy and resilience may be a successful strategy to improve child and parent wellbeing during the PICU stay and beyond. Systematic reviews have demonstrated that increasing parenting self-efficacy can reduce parental stress, and improve parent mental health, and child development [131,132]. 

Some early approaches to supporting parents during this challenging time have shown benefits to both parental and child outcomes after PICU. Anticipatory guidance and education consist of the information that clinicians give families about what they should expect during their child’s admission, how they can support their child’s development, how they can engage in care practices and advocacy for their child, and the importance of ongoing communication with the PICU staff [133]. Gains in maternal knowledge and health locus of control, increased frequency of cognitively stimulating activities and affection, and interaction between mother and child are effects of anticipatory education that can also be provided to the parents around periods of transition including at discharge [134,135]. 

Evidence suggests that psychosocial support during hospitalization, including mindfulness [136,137,138] and education to improve problem-based coping strategies [139,140] and mental health literacy [141,142], is associated with improved psychological symptoms and post-traumatic stress symptoms [143,144], in addition to child externalizing and internalizing behaviors [70,145]. PICU parent interventions encouraging soothing touch and talk while their child undergoes invasive procedures [146], increasing parents’ involvement in decision-making regarding their child’s care [147], and combinations of psychoeducation during admission with follow-up post-discharge [148], have reduced physiological arousal in children, and resulted in better psychological outcomes in both parents and children post-discharge. More intensive intervention approaches stretching beyond PICU admission are scarce, but one educational-behavioral intervention program called Creating Opportunities for Parent Empowerment (COPE) aimed at parents of young children was designed to begin shortly after admission to the PICU and extend to 2–3 days post hospital discharge. The program reduced parent stress during PICU admission and psychological sequelae after hospitalization and improved emotional and behavioral outcomes in children 12 months post-discharge [126,143]. A new, post-discharge, targeted, screen-and-treat, early intervention for reducing posttraumatic distress in young children screening high-risk of post-traumatic stress symptoms following PICU admission is currently undergoing a feasibility trial [149]. While the evidence for parental support and education programs is emerging, robust systems addressing the needs of carers and children from all circumstances re-quire validation and consistent implementation to avoid ongoing harm to children’s brains.

### 4.4. Early Screening and Intervention

Screening and assessment for vulnerabilities, morbidities and quality of life, and this information to guide patient care, has shown significant improvements in child outcomes and care indicators [150]. The goal of early screening is to provide early intervention where required, to prevent or minimize impairments, and is essential to young children’s optimal language, cognitive, motor, and socioemotional development, and successful academic performance [150,151]. To particularly advance our understanding of how PICU-related interventions, as well as changes in care, impact on person-centered outcomes, a commitment to obtaining baseline assessment as well as long term screening is urgently needed. This could include assessment of social determinants, ACEs, child development and parent mental health and functioning, to identify at-risk children and families on admission to the PICU.

A variety of post-PICU screening and assessments have been previously discussed including programs empowering careers with knowledge [152]. Follow-up can vary depending on who is assessed, when they are assessed, what measurement tools are used and how early interventions are engaged with this information. These decisions can only be made by local teams based on funding, capacity and capability. Prior to any decisions about the provision of early screening or assessment, clinical staff and parents require education on the potential impact of critical illness on the child and family, to ensure early conversations about preventative measures and escalation of concerns. Consistent messaging on the importance of long-term development is essential for uptake of strategies such as TIC and PICU Liberation.

Many acute care settings are currently not funded to invest in follow-up services or networked with relevant interdisciplinary teams to obtain meaningful outcomes for all children discharged from PICU. Changing this will require investment in new ways of providing these service solutions [152]. Intensive care clinicians could partner with allied health and developmental pediatricians as well as primary care providers to ensure tailored and meaningful follow-up and care, including the use of patient reported outcome measures [153,154,155,156,157]. Engaging with pediatric colleagues and primary healthcare providers to assist with follow-up support will require a review of the essential elements of PICU discharge summaries and communication to ensure a clear understanding of the risks of post-intensive care syndrome to the child and family, not just a summary of medical treatment provided [158]. Consideration of integrative systems that allow meaningful follow-up as required, based on the child’s risk profile and testing are crucial to not only record the currently under recognized health care burden, but also allow early intervention and improved outcomes.

## 5. Embracing Neuroscience in the PICU

Collaboration and communication are essential to enable parents and clinical staff to interact with young children who have experienced a critical illness successfully, and to inform caregivers about the influence they can have on their child’s neurodevelopment by using language and interactional strategies that are developmentally supportive [159,160,161]. Similar principles apply to all neurodevelopmental stages, not just infants, and the interdisciplinary team can both assess and guide care principles, interventions, and partner with carers to ensure seamless education and training as well as inclusion in bedside cares. Bedside techniques that support responsive relationships, including modelling empathy and active listening, encourage protective factors and build on family strengths [162]. Parents can be supported to help strengthen core life skills including parenting, self-regulation, and executive function [163]. Elucidating sources of stress for parents and children including stressors at home, e.g., finances, food security, and domestic violence, can help inform strategies to reduce stress and improve coping [164,165,166].

Over two decades, the Alberta Family Wellness Institute (AFWI) has provided the groundwork to bridge the gap between neuroscientific principles and brain vulnerability and how policy and practice can respond with impact. In collaboration with researchers from the Harvard Centre and the National Scientific Council on the Developing Child, the “Core Story of Brain Development” was developed. This serves as a tool for education and establishes a common language and recognition of how brain development is linked to later health outcomes. The AFWI website offers free Brain Story Certification, which has become a major resource for individuals and organizations (e.g., through mandatory education) to become change agents and develop and apply knowledge about brain science, child development to impact long-term health outcomes [167,168]. Through the Thriving Queensland Kids Partnership (TQKP) and Australia Research Alliance for Children and Youth (ARACY), Australia is now exploring how best to implement Brain Story Certification to change focus and behaviors, taking a human development and systems approach to improve policies, programs, and practices. Disrupting disadvantage and adversity and building and protecting healthy brain development starts with the individual, therefore it is recommended that PICU staff be supported to complete the Brain Story training. Further, PICUs need to consider making similar courses part of the mandatory training for all staff.

Health professionals have a critical role in developing and implementing workable, sustainable, and context-specific solutions. They not only advocate for the clinical interface but also for the larger scale-up of policies and interventions that support early childhood development. As clinicians who care for vulnerable children and their families, it is time for us to be intentional. We can help transform our knowledge about early childhood development into interventions that support children and families; this must be a PICU priority and placed at the top of our agendas. Using the guidance of the United Nations Children’s Fund (UNICEF) in ‘Early Moments Matter’ for every child [169], we can:Develop and invest in care and services that give young children the best start in life. This might include mandatory TIC training for all staff and carers.Make family-friendly early childhood development policies a PICU priority. This might include open visitation, parental presence during procedures and rounds, and neurodevelopmental cares.Collect data on essential indicators of early childhood development and track progress. This might include social determinants, adverse childhood events, pre-morbid child development, pre-PICU parent functioning, and post-PICU outcomes.Provide leadership for early childhood development programs. This might include Brain Story Certification, and the embedding of core brain story concepts into curriculum.Continue to advocate for early childhood development services that families can access. This might include parenting programs, care coordination, post-PICU child and family psychosocial interventions, and PICU follow-up services.

## 6. Conclusions and Future Directions

Brain science provides a pathway to healthy growth and development for children, young people, and families today and for generations to come. Most children do survive their critical illness or injury yet are exposed to environments and early experiences that require urgent attention, as they can contribute to negative trajectories and children’s ability to achieve their best possible potential [170]. Hospitalization and illness, as early adverse experiences that can impact the brain architecture of young children, require clinicians to simultaneously sustain best practice interventions and care that reduce child mortality while maintaining focus on brain vulnerability and development. Child development and child survival should be seen as complementary rather than as two opposing objectives. Whilst systems to recognize and respond to acute, severe and chronic illnesses continue to improve mortality, a focus shift can impact overall child and parent health and wellbeing by minimizing negative stressors, increasing positive experiences and relationships, and supporting the development of resilience and coping skills. By embracing the principles of neuroscience and brain development in PICU, small investments in the child and family at the bedside will have positive impacts that can be amplified exponentially.

The fundamental concepts of neuroscience and behavioral science help explain why a prosperous and sustainable society depends on the healthy development of children. Early intervention can prevent the consequences of adversity; with research showing that delayed interventions can be less successful [171]. While there is no identified ideal window for intervention, neuroscience principles suggest that providing supportive conditions for early childhood development is more effective and less costly than attempting to address the consequences of early adversity later [172]. PICU clinicians are ideally placed to not only intervene with life-saving therapies, but facilitate safe and strong relationships and environments and provide timely interventions to buffer the stressors of critical illness and hospitalization

## Figures and Tables

**Figure 1 children-09-01938-f001:**
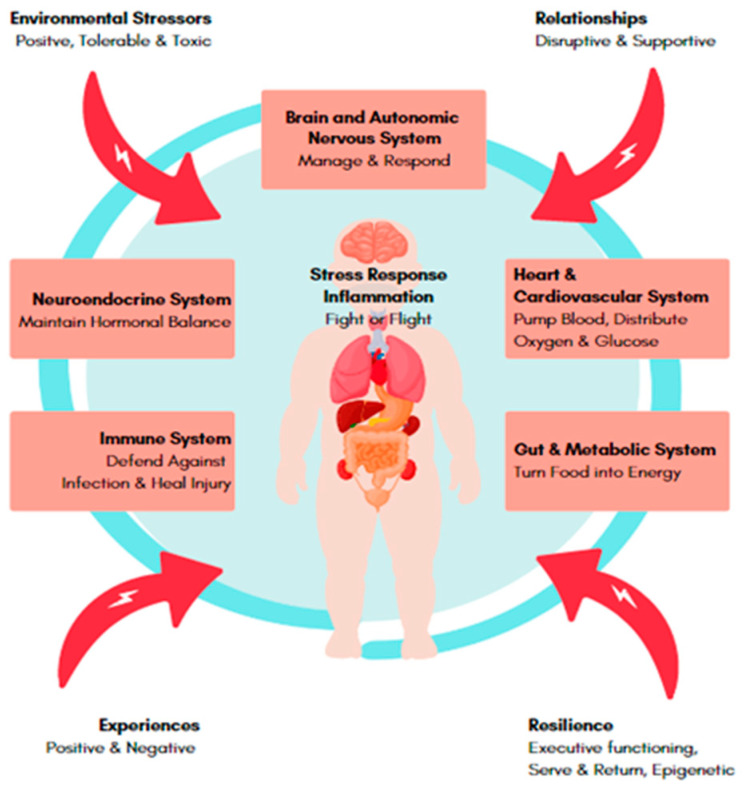
Impact of Stressors on Biological Systems.

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
