# Peer review of "Brain-Directed Care: Why Neuroscience Principles Direct PICU Management beyond the ABCs"

_children, 2022, doi:10.3390/children9121938_

Round 1

Reviewer 1 Report

Reviewer comments

 This is an interesting Review article on Brain directed care: Why neuroscience principles direct PICU 3 management beyond the ABCs with accepts for publication.

Comments

1-      The English used correct and readable.

2-      The work had a significant contribution to the field.

3-      The work was well organized and comprehensively described.

 4-      There were appropriate and adequate references to related and previous work.

Author Response

Many thanks for your positive feedback. We are glad that you enjoyed reading it.

Reviewer 2 Report

Dear authors, 

In the article is explained how the treatment of small children in PICU affects the developing brain and emphasizes the importance and possibilities of early interventions, to prevent neurological and psychological consequences. The authors explain in detail the latest scientific theses about the numerous influences on brain development and how a stressful stay in the PICU affects the short-term and long-term outcome of the child and the child's family. The importance of early interventions by health personnel is emphasized.

The article is very interesting and important for all health care workers in pediatric intensive care units: important suggestions related to this topic are offered. The topic is well explained and argued.

The article is well written, the text is clear, interesting, and easy to read.

The conclusions are clear.

I have just few suggestions:

1.    In Line 256 the abbreviation “TIP” I assume is for Trauma informed care? “Trauma informed care” is used in line 237, so an abbreviation should be given after it. After that, state the full name or abbreviation in the text.

2. In line 338 the abbreviation SES should be explained.

3. In line 356 the abbreviation PTSS should be explained

4. the references numbers in the text should be moved to the end of the sentence.

Author Response

Many thanks for your positive review and constructive feedback.

Thank you for bringing these errors and inconsistencies to our attention. We have amended the manuscript and provide it now with tracked changes.

Regarding your comment about the references, we have crossed checked with the author guidelines and papers already published by this journal and we feel that our referencing is appropriate. We have however, gone through the manuscript and amended the references to minimize mid-sentence referencing except where several discrete concepts are explored. Could the copy editor please advise of specific changes that need to be made and we will amend accordingly.